# New Structures, Spectrometric Quantification, and Inhibitory Properties of Cardenolides from *Asclepias curassavica* Seeds

**DOI:** 10.3390/molecules28010105

**Published:** 2022-12-23

**Authors:** Paola Rubiano-Buitrago, Shrikant Pradhan, Christian Paetz, Hannah M. Rowland

**Affiliations:** 1Research Group Predators and Toxic Prey, Max Planck Institute for Chemical Ecology, Hans Knöll Straße 8, 07745 Jena, Germany; 2Research Group Biosynthesis/NMR, Max Planck Institute for Chemical Ecology, Hans Knöll Straße 8, 07745 Jena, Germany

**Keywords:** Na^+^/K^+^ ATPase activity, Apocynaceae, toxicity, secondary metabolites, phytochemistry

## Abstract

Cardiac glycosides are a large class of secondary metabolites found in plants. In the genus *Asclepias*, cardenolides in milkweed plants have an established role in plant–herbivore and predator–prey interactions, based on their ability to inhibit the membrane-bound Na^+^/K^+^-ATPase enzyme. Milkweed seeds are eaten by specialist lygaeid bugs, which are the most cardenolide-tolerant insects known. These insects likely impose natural selection for the repeated derivatisation of cardenolides. A first step in investigating this hypothesis is to conduct a phytochemical profiling of the cardenolides in the seeds. Here, we report the concentrations of 10 purified cardenolides from the seeds of *Asclepias curassavica*. We report the structures of new compounds: 3-*O*-*β*-allopyranosyl coroglaucigenin (**1**), 3-[4′-*O*-*β*-glucopyranosyl-*β*-allopyranosyl] coroglaucigenin (**2**), 3′-*O*-*β*-glucopyranosyl-15-*β*-hydroxycalotropin (**3**), and 3-*O*-*β*-glucopyranosyl-12-*β*-hydroxyl coroglaucigenin (**4**), as well as six previously reported cardenolides (**5**–**10**). We test the in vitro inhibition of these compounds on the sensitive porcine Na^+^/K^+^-ATPase. The least inhibitory compound was also the most abundant in the seeds—4′-*O*-*β*-glucopyranosyl frugoside (**5**). Gofruside (**9**) was the most inhibitory. We found no direct correlation between the number of glycosides/sugar moieties in a cardenolide and its inhibitory effect. Our results enhance the literature on cardenolide diversity and concentration among tissues eaten by insects and provide an opportunity to uncover potential evolutionary relationships between tissue-specific defense expression and insect adaptations in plant–herbivore interactions.

## 1. Introduction

Plants produce a range of low molecular weight organic compounds, some of which are not involved in the ‘primary’ functions of plants but which mediate plant–environment interactions—known as secondary metabolites (or natural products) [1]. Most plant secondary metabolites have evolved to defend plants against insects and other natural enemies [2,3]. That does not mean, however, that all secondary compounds have a defensive function, and criteria for determining this are still not fully developed [1,4]. Even in some of the most well-studied systems, the structures and functions of these metabolites are still undescribed [5]. Therefore, testing the biological activity of secondary metabolites in a plant and, if they are active, whether they are of evolutionary and ecological significance is important for understanding the mechanisms and function of chemodiversity [5].

Milkweed plants in the genus *Asclepias* (Apocynaceae) are optimal candidates to investigate chemodiversity and secondary metabolite activity because their defenses of cardenolides and latex [6,7,8] vary among species [9,10] and among plant parts [11,12] and because cardenolides have a specific physiological target, the transmembrane protein Na^+^, K^+^−ATPase (NKA [3,4,5,6,9,11,13,14]), which can be tested in vitro [15]. Cardenolides are toxic because they bind to the extracellular surface of the NKA and, when bound, block this ion pump, leading to the breakdown of membrane potentials and cell homeostasis with potentially fatal effects [7,16,17,18]. Specialist herbivores, such as the monarch butterfly (*Danaus plexippus*), which feeds on the foliage of milkweed plants [19,20,21,22,23], and the large milkweed bug (*Oncopeltus fasciatus*), which feeds on the seeds [24,25,26,27], have reduced sensitivity to cardenolides through the evolution of amino acid substitutions in the NKA, preventing the cardenolides from binding [15,28,29,30,31,32,33]. The resulting arms race is thought to drive the diversity of plant secondary metabolites and insect herbivores [4].

Cardenolide diversity ranges up to 30 compounds in a single plant [7,34]. Plants show longitudinal trends in these natural products, with implications for the survival of herbivores [35]. Cardenolides vary in polarity [9], resulting in different rates of absorption through the gut of animals [36], with non-polar compounds being more readily absorbed than polar ones [37]. Cardenolides also vary in the structural sugar groups (the ‘glycoside’ of cardiac glycosides) that conjugate to the core aglycone steroidal structures of cardenolides, and this can alter the chemical properties of the molecule [38,39]. Cardenolides can also have reactive moieties, such as aldehydes that form H-bonds between the molecule and the NKA [39,40]. Sugars in cyclic bridges, such as a dioxane ring, are also highly resistant to acid hydrolysis [39]. Not all cardenolides, then, are predicted or shown to be equally toxic to herbivores [11,39].

Recently, Agrawal et al. (2022) tested the inhibitory capacity of a subset of purified cardenolides from *Asclepias syriaca* on the Na^+^/K^+^-ATPase. There was little variation among compounds in inhibition of an unadapted Na^+^/K^+^-ATPase, but impacts on that of monarchs and *Oncopeltus* varied significantly [41]. Here, we focus instead on *Asclepias curassavica* which, although native to the Caribbean area, is cultivated widely as an ornamental plant, and it is now found in numerous semitropical areas [6]. *A. curassavica* is understudied in its interactions with specialist herbivores in comparison with other species, despite being a critical hostplant worldwide [19]. The seeds of *A. curassavica* have received less attention than its foliage [40,42,43,44] but are subject to selection by seed herbivores [11]. The seeds of *A. curassavica* have one detailed phytochemical profiling by Abe et al [45] (though, see [11,46]). In that study, fourteen compounds were isolated, several of which have only been found in the seeds of *A. curassavica*, in contrast to some cardenolides that are present in foliage, latex, and roots [45]. Our goal was to describe the structures, concentrations, and activity of these metabolites.

## 2. Results and Discussion

### 2.1. Isolation and Structure Elucidation

For the isolation of cardenolides, 264 g of vacuum-dried *A. curassavica* seeds were ground and extracted exhaustively using water and MeOH as solvents. The crude extracts were subjected to HPLC-HRMS analyses. (Appendix A). Extracts containing chromatographic peaks with a cardenolide-like spectrum were selected for further purification. The cardenolide characteristics in the spectrometric data involved identifying values of a neutral loss that correspond to sugars in cardenolides (e.g., 162.05 Da- possible glucose and 146.05 Da-possible methyl allose). The spectrum must also contain high intensity fragments between *m*/*z* 353.20–391.25, tentatively corresponding to an unsaturated triterpene [47,48]. After the putative structural assignment based on HRMS data, the isolated compounds were examined by means of NMR spectroscopy. We identified six known cardenolides: 4′-*O*-*β*-glucopyranosyl frugoside (**5**) [49], 4′-*O*-*β*-glucopyranosyl gofruside (**6**) [50], 3′-*O*-*β*-glucopyranosyl calotropin (**7**) [45], frugoside (**8**) [50], gofruside (**9**) [51], and 16*α*-hydroxycalotropin (**10**) [45] (Figure 1). In addition, we isolated four new cardenolides (Figure 1): 3-*O*-*β*-allopyranosyl coroglaucigenin (**1**), 3-(4′-*O*-*β*-glucopyranosyl- *β*-allopyranosyl) coroglaucigenin (**2**), 3′-*O*-*β*-glucopyranosyl-15-*β*-hydroxycalotropin (**3**), and 3-*O*-*β*-glucopyranosyl-12-*β*-hydroxy-coroglaucigenin (**4**). All compounds were isolated as white solids. We quantified the concentration of all cardenolide in the seeds and tested the in vitro inhibitory capacity of each purified cardenolide on porcine NKA [15].

The molecular structures of *Asclepias* cardenolides have been intensively studied in the past [52,53,54]. On the basis of X-ray analysis, their principle structures have been determined: the triterpene scaffold of *Asclepias* cardenolides has the common feature of an α-orientation of the methine proton at C-5; therefore, the rings A and B of the scaffold are trans-fused [55]. Analysis of the NMR data led us to the conclusion that **1**, **2,** and **4** are coroglaucigenin-type molecules with a hydroxylation at C-19, while compound **3** is a calotropin derivative (Figure 2). We found compounds with one or two glycosylations, whereas in the previous study of the seeds, four cardenolides containing cellobiosyl units were reported [45].

Compound **1** has the molecular formula of C_29_H_44_O_10_, determined by the ion peak at *m*/*z* 553.3015 [M + H]^+^ (calculated for C_29_H_45_O_10_, *m*/*z* 553.3013). It is a coroglaucigenin derivative with one glycosylation, analogous to 4′-*O*-*β*-glucopyranosyl frugoside (**5**) [45]. However, the sugar moiety bound to the sterol at position C-3 shows, unlike for many cardenolides reported in *Asclepias*, an oxidized methylene at C-6′. (Appendix A)

Further analysis of the glycosyl relative configuration revealed that the carbinolic protons H-1′and H-3′ are both in equatorial position. Proton H-3′ is furthermore in syn-periplanar position to H-2′ and H-4′. We therefore assumed an *α*-oriented hydroxyl function in position C-3′ (Figure 3). This stereochemistry is characteristic for allose, the C-3′ epimer of glucose [56]. Compound **1** is, accordingly, 3-*O*-*β*-allopyranosyl coroglaucigenin (see Appendix A for the IUPAC name).

Compound **2** has the molecular formula C_35_H_54_O_15_, determined by the ion peak at *m*/*z* 715.3549 [M + H]^+^ (calcd for C_35_H_55_O_15_, 715.3535) (Appendix A). The structure is similar to compound **1** but shows two signals in the ^1^H NMR spectrum at *δ*_H_ 4.87 (*δ*_C_ 97.8, H-1′) and *δ*_H_ 4.56 (*δ*_C_ 103.6, H-1″) that we assigned to anomeric glycosyl positions. It suggested a glycosyl chain, where the first sugar was again an allose. For the second glycosyl moiety, however, the multiplicity of H-2′ and H-3′, both dd multiplicities with a large coupling constant (^3^*J*_HH_ > 9 Hz), revealed an anti-periplanar arrangement, consistent with the *β*-OH orientation at C-3′. This is characteristic for a glucosyl rest and, accordingly, compound **2** is 4′-*O*-*β*-glucopyranosyl-3-*O*-*β*-allopyranosyl coroglaucigenin (see Appendix A for the IUPAC name).

Compound **3** is a calotropin-type cardenolide with a molecular formula C_35_H_48_O_14_, determined by the ion peak at *m*/*z* 693.3113 [M –H_2_O + H]^+^ (calcd for C_35_H_49_O_14_, 693.3117) (Appendix A). This compound is similar to compound **7**, with the only difference being a hydroxylation at C-15. Analysis of the ^1^H-^1^H ROESY data showed that H-17 and H-15 are in syn-periplanar orientation. Given the absolute configuration of H-17 according to biosynthetic considerations, we assign H-15 as 15R [54]. We determine compound **3** as 3′-*O*-*β*-glucopyranosyl-15*β*-hydroxycalotropin (see Appendix A for the IUPAC name).

Compound **4** has a molecular formula of C_29_H_44_O_11_, determined by the ion peak at *m*/*z* 569.2970 [M + H]^+^ (calcd for C_29_H_45_O_11_, 569.2956). It is of the coroglaucigenin type but, in this case, with a carbinolic proton resonating at *δ*_H_ 3.32 (*δ*_C_ 74,7, H-12). We defined the stereocenter of this oxydized methine as (R) from ^1^H-^1^H ROESY correlations between H-1*α*↔H-9↔H-12↔H-15*α*↔H-17. Furthermore, the large coupling between H-12 H_2_-11*β* (^3^*J*_HH_ = 12.2 Hz) was consistent with their trans-periplanar orientation. The sugar moiety was identified as glucose by the large coupling constants and the ^1^H-^1^H ROESY correlations between H-1′ and H-5′. We therefore describe the compound as 3-*O*-*β*-glucopyranosyl-12*β*-hydroxy coroglaucigenin (see Appendix A for the IUPAC name).

### 2.2. Quantification of Cardenolides

We quantified compounds **1**–**10** in mg of compound per gram of seeds (dry weight; Figure 4 and Appendix A). We used each compound as a standard for their corresponding calibration curve (Appendix A). Compound **5**, 4′-*O*-*β*-glucopyranosyl frugoside, is the most abundant cardenolide, with 4.5 mg/g of seeds, approximately two times more than 4′-*O*-*β*-glucopyranosyl gofruside **6** (2.06 mg/g). The other eight cardenolides are present in amounts below 1 mg/g. Compounds **1** and **4** are the least abundant at 0.01 and 0.004 mg/g, respectively.

The range of retention times of the isolated compounds was 19 to 30 min (Appendix A, for chromatography conditions see Section 3.1). The minor compounds **1**–**4** and **10** have a higher polarity than the more abundant compounds (Figure 4). Recently, López-Goldar et al. [11] reported a predominance of more polar compounds in the seed extracts of *A. curassavica*. Allomethylose and deoxy-allomethylose are present as sugar moieties in the abundant cardenolides **5**–**10**, whereas allose and glucose were found in the minor compounds **1**, **2**, and **4**. This finding may provide clues regarding the biosynthetic pathways of the rare sugars in *Asclepias*, where allosyl cardenolides could be intermediates of interest.

### 2.3. Na^+^/K^+^ ATPase (NKA) Inhibitory Activity

We tested the inhibitory capacity of the new compounds **1** and **2** and the known cardenolides 5–10 against porcine NKA. We used an in vitro assay to determine the IC_50_ of each compound and used ouabain as a reference.

There was a significant variation in inhibition of an unadapted NKA from *Sus domesticus* (Appendix A). The IC_50_ values ranged from 10^−6^ to 10^−8^ M; this is consistent with the sensitivity for cardenolide inhibition expected from *S. domesticus* NKA [19,39,57]. Gofruside **9** was the most inhibitory compound tested (IC_50_ = 9.653 × 10^−8^ M). The least inhibitory was 16*α*-hydroxycalotropin 10, with an IC_50_ of 3.667 × 10^−6^ M (Figure 5).

Compounds **1** and **8** are coroglaucigenin derivatives with allosyl and allomethylosyl substitution, respectively, but they did not differ in their inhibition properties (Appendix A). Compounds **8** and **9** share the same glycosylation, but **9** shows a different oxidation state of C-18 (alcohol vs. aldehyde). The higher inhibition of **9** compared with **8** suggests that the aldehyde is the main reason for the difference. This can be due to the high reactivity of **9** towards biomolecules. However, 16*α*-hydroxycalotropin **10** also contains an aldehyde but has the lowest inhibitory capacity. It has been reported that calotropin, the 16-deoxy aglycone of **10**, has an IC_50_ of 2.7 × 10^−7^ M (log_10_: −6.56) against porcine NKA [19]. This difference in inhibition may also be attributed to the fact that in **10,** the 16*α*-hydroxylation interferes with binding to the biological target. Molecular docking analyses are required for a better understanding.

We found a reduced inhibitory potential when a glucosylation of cardenolides occurred. The compounds **8** and **5** (coroglaucigenin-type) and **9** and **6** (corotoxigenin-type) differ only in the glucosylation, the aglycones having the higher inhibition potential. However, we also found that compounds **1** and **2**, which differ in the presence of glucose as a second sugar unit, have no significant difference in their inhibitory potential when compared with one another. An explanation for this could be the increased bulkiness of the molecule caused by the higher degree of glycosylation in both cases, which hinders the access to the active site of the NKA. In addition, in this case, molecular docking analyses would be needed to unravel the impact of the glycone moieties in the inhibition of NKAs.

Overall, our Na^+^/K^+^ ATPase inhibition results by cardenolides from *Asclepias* seeds are in line with a previous comparison of structural characteristics and NKA inhibition by Petschenka et al. [39], who described a differential response in the inhibition of vertebrate NKA by cardenolides with the same aglycons but different glycosylations.

Phytochemical diversity, like that described here, is linked to herbivore community in several systems [58,59,60], but disentangling concentration and inhibitory potency of phytochemicals is challenging and may be related to the costs of producing phytochemicals for the plant and differences in toxicity against herbivores with specific tolerance mechanisms [11]. In our future work we will seek to identify the selective pressures that lead to the differential investment in the compounds, given their different inhibitory effects against porcine NKA, and determine whether these effects vary depending on different cardenolide tolerances.

## 3. Materials and Methods

### 3.1. General Experimental Procedures

Nuclear magnetic resonance (NMR) spectra were recorded on a Bruker Avance III HD spectrometer (Bruker BioSpin GmbH, Rheinstetten, Germany) equipped with a cryoplatform and a 5 mm TCI CryoProbe, field strengths of ^1^H (500.13 MHz)/^13^C (125.76 MHz): 11.747 T. Spectrometer control and data processing was accomplished using Bruker TopSpin 3.6.1, and standard pulse programs as implemented in Bruker TopSpin 3.6.1. were used. Samples were measured in MeOH-*d*_3_ (99.5%) or D_2_O (99.9%), depending on solubility of the compounds. For compounds measured in MeOH- *d*_3_, the residual solvent signals were *δ*_H_ 3.31/*δ*_C_ 49.15. For measurements in D_2_O, all chemical shifts were left uncorrected after carefully tuning and matching the NMR instrument. High performance liquid chromatography coupled to high resolution mass spectrometry (HPLC-HRMS) analyses were performed on an Agilent 1260 Infinity, using a reversed-phase column Agilent Zorbax RP-18e (3.5 μm particle size, 3 × 150 mm). The mobile phase consisted of acetonitrile (ACN, supplied with 0.1% formic acid, FA, Carl Roth GmbH, Karlsruhe, Germany) and water (HPLC grade, 0.1% FA, deionized with a Merck Millipore Milli-Q A10, Merck KgA, Darmstadt, Germany). An elution gradient was used as follows: starting with ACN/H_2_O (5:95) for 5 min, then to ACN/H_2_O (45:55) for 25 min and maintained for 10 additional min. Later, it was set back to ACN/H_2_O (5:95) for 2 min and maintained for another 3 min, a total time of 45 min. For calibration curves and quantification of each compound, an Agilent Poroshell 120 column (2.7 µm particle size, 4.6 × 50 mm) was used. The elution gradient started with ACN/H_2_O (5:95) for 1 min, then to ACN/H_2_O (95:5) for 8 min, which was maintained for 2 min. Later, it was set back to ACN/H_2_O (5:95) for 1 min. The gradient had a total length of 12 min.

High resolution mass spectra were recorded on a Bruker Compact OTOF spectrometer (Bruker Daltonics GmbH, Bremen, Germany). Electrospray ionization (ESI) in positive ion mode was used for the analysis in full scan and auto MS/MS modes, scanning masses from *m*/*z* 50–1300. Capillary voltage was set at 4500 V, charging electrode at 2000 V, and corona current at 0 nA; nebulizer pressure gas was set at 1.8 bar, drying gas temperature at 220 °C, and drying gas flow at 9.0 L/min. Sodium formate adducts were used for internal calibration with a Quadratic + HPC mode. Bruker Compass ver.1.9 (OTOF Control ver.5.1.107 and HyStar 4.1.31.1) was used for data acquisition and instrument control, and Bruker DataAnalysis ver. 5.1.201 was used for data processing.

Reversed-phase MPLC separations were carried out on a Biotage Isolera One (Biotage SB, Uppsala, Sweden) using a Biotage Sfär C18 D—Duo 100 Å 30 µm 120 g column. A linear gradient, using a mobile phase consisting of MeOH (supplied with 0.1% FA, Carl Roth GmbH) and water (0.1% FA) was used, with a flow rate of 50 mL/min, and UV detection was carried out at 218 nm. For separations on Sephadex LH-20 (VWR GmbH, Dresden, Germany), a column containing 44 g of sorbent was used and eluted with the Isolera MPLC equipment, using water as the mobile phase. MPLC separations on MCI gel CHP20P (Merck KgA, Darmstadt, Germany) were carried out using a linear MeOH-water gradient on the Biotage Isolera equipment. Semi-preparative HPLC separations were carried out on a Shimadzu Prominence HPLC System, consisting of an autosampler SIL-20AC, gradient pump LC-20AT, UV-Vis detector SPD-20A programed for detection at 220 nm and a fraction collector FRC-10A. For separations, isocratic elutions with MeOH-water mixtures were performed at a flow rate of 0.8 mL/min. A C-18 Nucleodur Isis column (4.6 × 250 mm, 5 μm particle size, from Macherey-Nagel, Düren, Germany) was used. The purity of the isolated compounds was calculated by ^1^H qNMR experiments, with ouabain as external standard (ERETIC).

### 3.2. Extraction and Isolation

*A. curassavica* seeds were purchased from Jelitto Perennial Seeds (Art. No.: AA974). We ground 264 g of desiccated seeds to powder and extracted with 100% water, water/MeOH (1:1), and later 100% MeOH. Each of the extracts was dried in vacuum, then re-suspended in MeOH, left at −20 °C overnight, and filtered through paper (Whatman grade 50, 185 mm diameter). We centrifuged the filtrate for 10 min at 13,200 rpm to remove remaining particles. The filtrates were pooled together, and the solvent was removed in vacuum by rotary evaporation at 40 °C. We obtained 11.0 g of crude extract. We separated the crude extract using reversed-phase (RP-18) silica gel by MPLC, obtaining 13 fractions (See Section 3.1). We first targeted the masses of the cardenolides previously isolated from *A. curassavica* seeds by HPLC-HRMS and analyzed fragmentation patterns that indicated the presence of steroidal glycosides [45,47,48]. We detected the presence of cardenolides in fractions 6 to 11. We used fraction 10 (F10) to isolate the cardenolides described here.

F10 (506.5 mg) was subjected to an MCI CHP20P gel column eluted with MeOH, giving five fractions. F10.2 and F10.3 were further separated on Sephadex LH-20 using water as eluent, giving the fractions F10.2.1-4 and F10.3.1-5. Sub-fractions that contained the same cardenolides were pooled together according to the HPLC-HRMS analyses. The combined fractions were further separated by reversed-phase HPLC with isocratic elution (MN C-18 Isis, ACN/H_2_O (2:8)). From F10, we isolated glucopyranosyl frugoside (**5**, 88 mg, 98% purity), glucopyranosyl gofruside (**6**, 4.2 mg, 72% purity), and glucopyranosyl calotropin (**7**, 6.5 mg, 97% purity). Similarly, F11 (2.3 g) was separated (ACN/H_2_O, 25:75). We obtained frugoside (**8**, 153 mg, 98% purity) and gofruside (**9**, 49 mg, 98% purity).

F9 (77.7 mg), separated with a mobile phase of ACN/H_2_O (16:84), yielded compound **1** (3 mg, 75% purity). F6-8 (86.7, 39.6, and 48.6 mg, respectively) were separated in a similar way (ACN/H_2_O isocratic mobile phase: 12:88, 14:86, and 15:85, respectively) and produced compounds **2** (3.7 mg, 91% purity), **3** (1.1 mg, 58% purity), **4** (2.5 mg, 48% purity), and 16*α*-hydroxycalotropin (**10**, 4.4 mg, 58% purity)**.** The lower purity levels of compounds 1-4 were due to the persistent co-elution with several minor compounds. The isolated cardenolides, together with their chemical data, are listed below. For ^1^H NMR and ^13^C NMR data, see Table 1 and Appendix A.

3-*O*-*β*-allopyranosyl coroglaucigenin (**1**) [*α*]_D_^25^ +3.4510 +/− 5.0554 S.D. (*c* 0.14, H_2_O), UV (ACN/H_2_O): 220 nm; HRESIMS *m*/*z* 553.3015 [M + H]^+^ (calcd for C_29_H_45_O_10,_ 553.3007, *Δ* 1.4 ppm).

4′-*O*-*β*-glucopyranosyl-3-*O*-*β*-D-allopyranosyl coroglaucigenin (**2**) [*α*]_D_^25^ +112.5669 +/− 5.8027 S.D. (*c* 0.12, H_2_O), UV (ACN/H_2_O): 220 nm; HRESIMS *m*/*z* 715.3549 [M + H]^+^ (calcd for C_35_H_55_O_15_, 715.3535 *Δ* 1.9 ppm).

3′-*O*-*β*-glucopyranosyl 16*β*-hydroxycalotropin (**3**) [*α*]_D_^25^ +33.6123 +/− 9.5351 (*c* 0.08, H_2_O), UV (ACN/H_2_O): 218 nm; HRESIMS *m*/*z* 693.3113 [M –H_2_O + H]^+^ (calcd for C_35_H_49_O_14_, 693.3117, *Δ* 0.6 ppm).

3-*O*-*β*-glucopyranosyl 12*β*-hydroxy coroglaucigenin (**4**) [*α*]_D_^25^ +55.5112 +/− 5.8428 (*c* 0.11, H_2_O), UV (ACN/H_2_O): 220 nm; HRESIMS *m*/*z* 569.2970 [M + H]^+^ (calcd for C_29_H_45_O_11_, 569.2956, *Δ* 2.5 ppm).

4′-*O*-*β*-glucopyranosyl frugoside (**5**) white powder, UV (ACN/H_2_O): 220 nm; HRESIMS *m*/*z* 699.3593 [M + H]^+^ (calculated for C_35_H_55_O_14_, 699,.3586 *Δ* 1 ppm).

4′-*O*-*β*-glucopyranosyl gofruside (**6**) white powder, UV (ACN/H_2_O): 220 nm; HRESIMS *m*/*z* 697.3427 [M + H]^+^ (calcd for C_35_H_53_O_14,_ 697.3430, *Δ* 0.4 ppm).

3′-*O*-*β*-glucopyranosyl calotropin (**7**) white powder, UV (ACN/H_2_O): 220 nm; HRESIMS 67.3161 *m*/*z* [M –H_2_O + H]^+^ (calcd for C_35_H_49_O_13_, 677.3168, *Δ* 1 ppm).

Frugoside (**8**) white powder, UV (ACN/H_2_O): 220 nm; HRESIMS *m*/*z* 537.3059 [M + H]^+^ (calcd for C_29_H_49_O_9,_ 537.3058, *Δ* 0.2 ppm).

Gofruside (**9**) white powder, UV (ACN/H_2_O): 222 nm; HRESIMS *m*/*z* 535.2894 [M + H]^+^ (calcd for C_29_H_43_O_9_, 535.2902, *Δ* 1.5 ppm).

16*α*-hydroxycalotropin (**10**) white powder, UV (ACN/H_2_O): 220 nm; HRESIMS *m*/*z* 549.2706 [M + H]^+^ (calcd for C_29_H_41_O_10_ 549.2694, *Δ* 2 ppm).

### 3.3. Quantification of Cardenolides

We quantified the cardenolides isolated from the *A. curassavica* seeds using HPLC-HRMS with a linear calibration method. Each compound was diluted in order to obtain ten data points in a concentration range from 0 to 1 mg/mL, and each value was corrected according to the measured purity of the compound. The spectrometric data were obtained using the method described in Section 3.1. We extracted the ion chromatograms on the basis of the *m*/*z* of the most abundant peak for each cardenolide ([M+H]^+^ for compounds **1**–**6** and **8**–**10** and [M+H-H_2_O]^+^ for compound **7**). The area of the corresponding peak was calculated and used as a quantification parameter. The data points that were out of the linear range, especially at high concentration, were excluded. Linear regression for compounds **1**–**10** was calculated with concentration and peak area as variables (Appendix A).

Analysis of the content of cardenolides **1**–**10** in the seeds was performed as follows: 6 g of *A. curassavica* seeds were collected from plants grown in the greenhouse of the MPI for Chemical Ecology in Jena, Germany (seeds purchased from Jelitto Perennial Seeds, Art. No.: AA974). Seed samples (three technical replicates, each of 2 g) were freeze-dried, ground, and exhaustively extracted with MeOH/H_2_O (1:1) three times and then twice more with MeOH. The extracts were pooled and filtered using grade 50 Whatman paper filter to remove particles. Lipophilic substances were removed by passing the filtrate through a MN HR-X 500 mg cartridge. The samples were then dried using N_2_ gas at 36 °C, obtaining 214.9 mg, 230.6 mg, and 218.9 mg, respectively, of raw extract. The HPLC-HRMS measurements were conducted as described above. From each technical replicate a solution of 2.6 mg/mL was prepared, and from each of these solutions, three injections of 5 μL were used for analysis, resulting in nine data points per compound. Data processing was accomplished as for the calibration curve, by extracting the ion chromatogram per compound and measuring the corresponding peak area. The concentration per injection was later expressed in mg of cardenolide per g of seed (dry weight).

### 3.4. Na^+^/K^+^ ATPase (NKA) Inhibitory Activity Assay

We quantified the NKA inhibitory activity of the isolated cardenolides using purified NKA from *S. domesticus* cerebral cortex (Sigma-Aldrich-A7510-5UN, Steinheim, Germany), following protocols standardized by Petschenka et al. [15]. Briefly, porcine NKA was diluted in di-water to a final assay concentration of 0.01 U/mL. The ATPase was exposed to exponentially decreasing concentrations from 10^−3^ to 10^−8^ M of ouabain (Sigma-Aldrich, O3125-1G, Steinheim, Germany) or the cardenolides **1**–**2** and **5**–**10**. There was an insufficient amount of compounds **3** and **4** to allow testing. We prepared a stock solution of each cardenolide in 100% DMSO, which was used for preparing the concentrations for the assay. The concentration of DMSO in the final solutions did not exceed 2% per well. The reaction mixture with NKA and cardenolides was incubated at 37 °C for 10 min, followed by addition of ATP (Sigma Aldrich, A9062-1G, Steinheim, Germany) and another incubation for 20 min. The NKA activity after cardenolide exposure was determined by quantification of inorganic phosphate released from enzymatically hydrolysed ATP. The ATP levels were measured by photometric determination, reading the absorbance at 700 nm with a microplate reader (BMG Clariostar, BMG Labtech, Germany). The absorbance of each reaction was corrected with the respective background, and the inhibition curves were plotted in R studio [61] with log_10_-transformed cardenolide concentration versus percent of uninhibited control, with top and bottom asymptotes set to 1 and 0, respectively [30,32,62]. For each cardenolide, we carried out three biological replicates, with two technical replicates, resulting in each data point being an average of six measurements. IC_50_ values for each cardenolide and ouabain were also calculated from the inhibition curves using R studio. We employed Bonferroni-adjusted significance tests for pairwise comparisons between the IC_50_ values for cardenolides **1**–**2** and **5**–**10** and ouabain. The isolated cardenolide purity was addressed in the preparation of the stock solutions for compounds **5** and **7**–**9**. For compounds **1, 2, 6**, and **10**, stock solutions were prepared without this correction, given the low amounts available and/or low purity; instead, the final IC_50_ concentrations values were adjusted with purity percent as the factor.

## 4. Conclusions

The re-examination of the seeds of *A. curassavica* allowed us to describe and quantify four new cardenolides (**1**–**4**). We also isolated six known compounds (**5**–**10**) and confirmed by spectrometric quantification that 4′-*O*-*β*-glucopyranosyl frugoside **5** is the most abundant cardenolide in the seeds. The examination of the isolated compounds against NKA enzymes from a sensitive vertebrate (*S. domesticus*) revealed that the most inhibitory cardenolide is gofruside **9,** and the least inhibitory are 4′-*O*-*β*-glucopyranosyl frugoside **5** and 16*α*-hydroxycalotropin **10**. Comparison of the IC_50_ values obtained with the structural characteristics of each cardenolide confirmed that glycosylation, when leading to higher polarity, corresponds directly to a decrease in toxic potential of the cardenolides. However, the structure of the glycosyl substituents and the degree of oxidation at position 19 (alcohol vs. aldehyde) in coroglaucigenin-type cardenolides can also influence inhibitory capacity. The chemical defense of *A. curassavica* seeds varies in quantity and NKA inhibition potential. The cardenolide profile described here may be the result of biosynthetic constraints that result in a high diversity of bioactive compounds, which increases the chances of achieving an effective defense against several herbivore pressures.

## Figures and Tables

**Figure 1 molecules-28-00105-f001:**
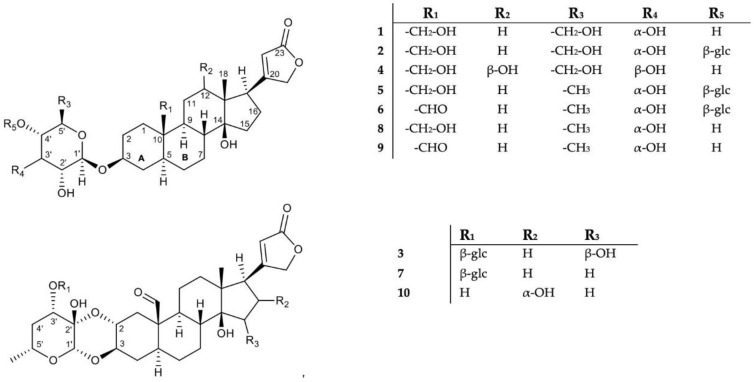
Structures of compounds **1–10** (glc, glucosyl).

**Figure 2 molecules-28-00105-f002:**
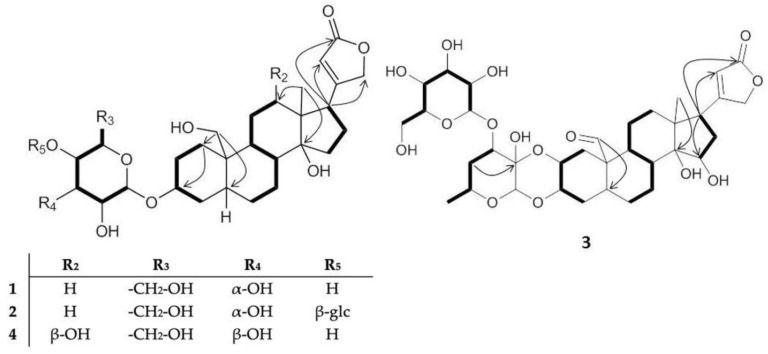
Important ^1^H-^1^H COSY (bold lines) and ^1^H-^13^C HMBC long-range correlations (arrows) for compounds **1–4** (glc, glucosyl).

**Figure 3 molecules-28-00105-f003:**
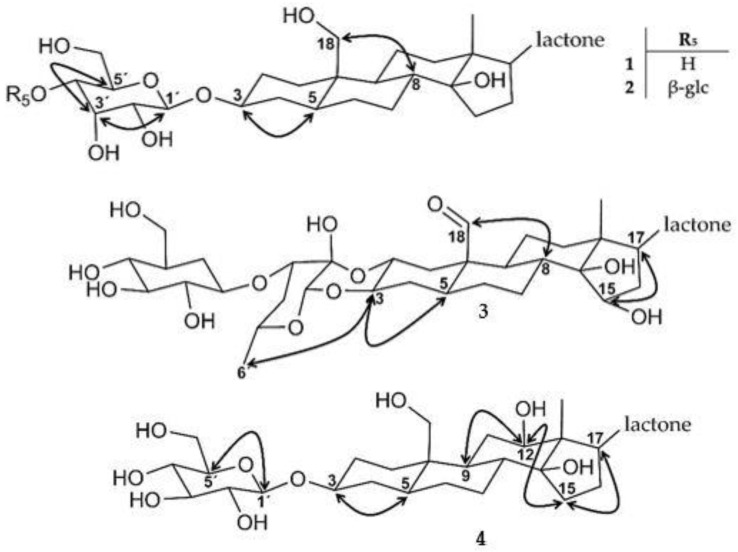
Important ^1^H-^1^H ROESY correlations (arrows) for compounds **1–4** (glc, glucosyl).

**Figure 4 molecules-28-00105-f004:**
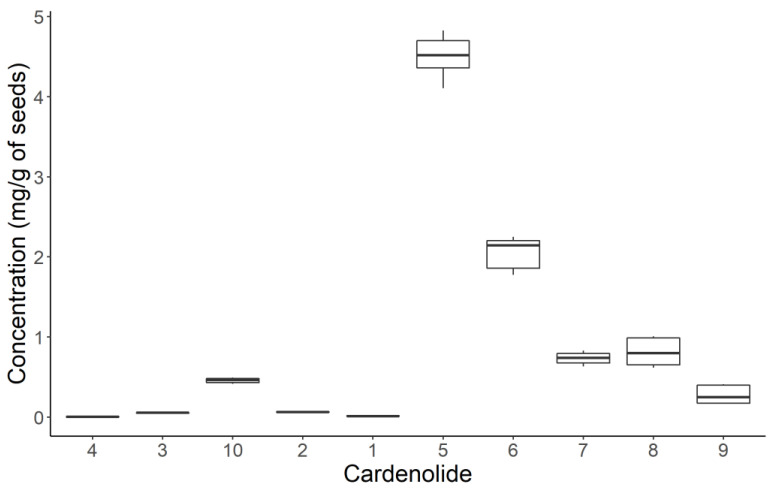
Quantification of the isolated cardenolides **1–10** from polar to non-polar (left to right) according to their retention time in reversed-phase chromatography. Boxplots show the median, interquartile range, and the whiskers represent the largest and smallest value within 1.5 times the 25^th^ and 75^th^ percentile.

**Figure 5 molecules-28-00105-f005:**
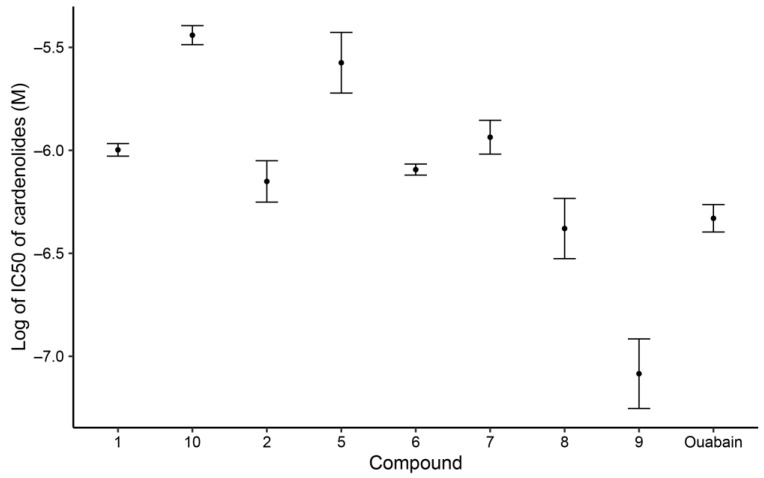
Mean + se IC_50_ values of isolated cardenolides 1, 2, and 5–10 and ouabain as reference.

**Table 1 molecules-28-00105-t001:** ^1^H- and (500.13 MHz) ^13^C-NMR data (125.76 MHz) of compounds **1**-**4** in D_2_O (*δ* ppm, rel. orientation, multiplicity^*^, (*J* [Hz])).

Position	1		2		3		4	
	*δ* _H_	*δ* _C_	*δ* _H_	*δ* _C_	*δ* _H_	*δ* _C_	*δ* _H_	*δ* _C_
1	2.15 *β* brd (13.4) 0.81 *α* dd (13.4, 11.1)	30.7 CH_2_	2.23 br, 0.89 br	30.6 CH_2_	2.45 *β* dd (12.6, 4.4) 1.19 *α* brt (12.6)	34.4 CH_2_	2.15 *α* dt (13.5, 3.1) 0.83 *α* brt (13.4)	30.5 CH_2_
2	1.85 brd, (11.1), 1.35 br	29.1 CH_2_	1.92 br, 1.43 br	29.2 CH_2_	3.91 *β* br	69.1 CH	1.86 br, 1.37 br	28.9 CH_2_
3	3.77 m	79.1 CH	3.85 m	79.1 CH	4.02 *α* td (10.6, 4.0)	72.1 CH	3.78 *α* m	79.0 CH
4	1.72 brd (12.2), 1.31 br	34.0 CH_2_	1.81 br, 1.41 br	34.0 CH_2_	1.76 *α* br, 1.31 *β* br	32.6 CH_2_	1.72 *α* br, 1.31 *β* q (12.2)	33.8 CH_2_
5	1.19 brt (11.5)	43.7 CH	1.28 br	43.8 CH	1.70 *α* br	42.2 CH	1.20 *α* br	43.8 CH
6	1.26 m, 1.26 m	27.3 CH_2_	1.34 m, 1.34 m	27.4 CH_2_	2.16 m, 1.57 m	26.1 CH_2_	1.25 m, 1.25 m	27.3 CH_2_
7	1.88 m, 1.07 m	26.9 CH_2_	1.96 m, 1.16 m	26.9 CH_2_	1.90 *β* m, 1.74 *α* m	26.9 CH_2_	1.86 m, 1.06 m	27.1 CH_2_
8	1.63 br	41.2 CH	1.71 br	41.3 CH	1.72 *β* br	41.8 CH	1.63 *β* br	40.6 CH
9	1.00 br	49.1 CH	1.08 br	48.9 CH	1.64 *α* br	46.9 CH	1.02 *α* brt (13.8)	45.3 CH
10	-	38.7 C	-	38.7 C	-	53.3 C	-	38.6 C
11	1.55 m, 1.33 m	22.5 CH_2_	1.64 m, 1.42 m	22.5 CH_2_	1.73 *α* m, 1.12 *β* m	21.5 CH_2_	1.76 *α* m, 1.46 *β* q (12.5)	30.6 CH_2_
12	1.44 m, 1.33 m	39.7 CH_2_	1.53 m, 1.42 m	39.7 CH_2_	1.53 m, 1.46 m	37.2 CH_2_	3.32 *α* dd (12.2, 1.7)	74.7 CH
13	-	49.8 C	-	49.5 C	-	48.3 C	-	55.8 C
14	-	86.3 C	-	85.7 C	-	82.1 C	-	86.4 C
15	2.09 m, 1.63 m	31.7 CH_2_	2.17 m, 1.72 m	31.8 CH_2_	4.66 *α* brd (8.4)	71.8 CH	1.88 *α* m, 1.67 *β* m	31.7 CH_2_
16	2.09 m, 1.73 m	26.5 CH_2_	2.19 m, 1.82 m	26.6 CH_2_	2.68 m, 1.66 m	36.0 CH_2_	2.10 m, 1.79 m	26.8 CH_2_
17	2.81 *α* br	50.2 CH	2.89 *α* br	50.2 CH	2.77 *α* dd (9.8, 4.9)	47.3 CH	3.20 br	45.4 CH
18	0.82 *β* s	15.2 CH_3_	0.91 *β* s	15.3 CH_3_	0.85 *β* s	16.1 CH_3_	0.73 *β* s	8.9 CH_3_
19	3.82, br 3.67 br	59.0 CH_2_	3.90 br, 3.76 br	59.0 CH_2_	10.10 s	213.1 CH	3.81 d (12.2) 3.68 d (12.2)	58.6 CH_2_
20	-	178.5 C	-	178.2 C	-	177.6 C	-	178.4 C
21	4.99 d (18.8) 4.93 d (18.8)	75.2 CH_2_	5.06 d (18.7) 5.00 d (18.7)	75.1 CH_2_	5.09 d (18.3) 5.02 d (18.3)	75.0 CH_2_	4.95 br, 4.95 br	75.1 CH_2_
22	5.89 s	115.9 CH	5.98 s	115.7 CH	6.02 s	116.2 CH	5.92 s	116.2 CH
23	-	179.3 C	-	178.9 C	-	178.2 C	-	178.5 C
1′	4.77 d (8.3)	98.0 CH	4.87 d (8.2)	97.8 CH	4.63 s	94.8 CH	4.51 *α* d (7.8)	100.3 CH
2′	3.32 dd (8.3, 3)	70.2 CH	3.43 dd (8.7, 2.7)	70.0 CH	-	91.6 C	3.14 *β* brt (8.7)	73.1 CH
3′	4.09 t (3)	71.3 CH	4.45 t (3.1)	70.9 CH	3.95 br	81.5 CH	3.39 *α* t (9.3)	75.8 CH
4′	3.53 dd (10, 3)	66.9 CH	3.78 dd (10, 2.7)	76.2 CH	2.13 *β* br, 1.71 *α* br	37.1 CH_2_	3.29 *β* t (9.3)	69.6 CH
5′	3.68 brddd (10.6, 1.5)	73.6 CH	3.88 brd (10.3, 2.2)	72.4 CH	3.82 *β* q (5.8)	68.7 CH	3.36 *α* ddd (9.3, 5.8, 1.7)	75.9 CH
6′	3.82 dd (12.0, 1.5) 3.61 dd, (12.0, 6.0)	61.2 CH_2_	3.90 brd (12.1) 3.76 dd (12.5, 4.2)	60.6 CH_2_	1.29 *α* d (6.3)	19.8 CH_3_	3.82 dd (1.7, 12.2) 3.63 dd (5.8, 12.2)	60.7 CH_2_
1”			4.56 d (7.9)	103.6 CH	4.64 d (7.7)	104 CH		
2”			3.32 brt (8.9)	73.2 CH	3.37 t (9.1)	73.4 CH		
3”			3.48 t (9.2)	75.7 CH	3.51 t (9.1)	75.5 CH		
4”			3.42 br	69.3 CH	3.43 br	69.4 CH		
5”			3.44 br	75.7 CH	3.44 br	75.8 CH		
6”			3.90,br, 3.76 br	60.6 CH_2_	3.90 dd (12.2, 1.5) 3.74 dd (12.2, 5.7)	60.2 CH_2_		

## Data Availability

Not applicable.

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
