# Peer review of "New Structures, Spectrometric Quantification, and Inhibitory Properties of Cardenolides from *Asclepias curassavica* Seeds"

_molecules, 2022, doi:10.3390/molecules28010105_

Round 1
Reviewer 1 Report
The manuscript “New structures, spectrometric quantification, and inhibitory properties of cardenolides from Asclepias curassavica seeds” reported the isolation and structural elucidation of four new 3-O-β-allopyranosyl coroglaucigenin (1), 3-[4’-O-β-glucopyranosyl-β-allopyranosyl] coroglaucigenin (2), 3’-O-β-glucopyranosyl-15-β-hydroxycalotropin (3), 3-O-β-glucopyranosyl-12-β-hydroxyl coroglaucigenin (4) together with six known cardenolides (5–10). The concentrations and in vitro inhibitory activity of isolated cardenolides on porcine Na+/K+-ATPase were reported
I recommend the manuscript will be accepted after minor revisions.
1. Characters α, β in itaclic. The authors should check the entire manuscript 2-4, 9
2. The Latin name of plant in italic (Line 96,108, 167, 170, 267, 311, 323).
3. Check the HRESIMS of new compounds 1-4. I found all the HRESIMS data are not fit with SI.
4. Typo errors: line 197: Aglycon, line 221: MeOH-d4, line 258: seeds (normal. Not in italic).
5. The authors check carefully the references : The names of journal should be written using abbreviations. (Ref. 10-12, 19-25, etc)
Author Response
We have carefully considered the reviewer’s suggestions, and produced a detailed description of how we have revised our manuscript (numbering of the comments is ours).
R1.1. Characters α, β in itaclic. The authors should check the entire manuscript.
Done
R1.2. The Latin name of plant in italic (Line 96,108, 167, 170, 267, 311, 323).
Done
R1.3. Check the HRESIMS of new compounds 1-4. I found all the HRESIMS data are not fit with
SI.
HRESIMS data in the manuscript has been corrected.
R1.4. Typo errors: line 197: Aglycon, line 221: MeOH-d4, line 258: seeds (normal. Not in italic)
Typos resolved. Note that measurements were carried out with MeOH-d3. Changes made to reflect this on Lines 227.
R1.5. The authors check carefully the references: The names of journal should be written using abbreviations. (Ref. 10-12, 19-25, etc)
All references have been checked and revised.

Reviewer 2 Report
Cardiac glycosides are a large class of secondary metabolites found in plants. In the genus Asclepias, cardenolides have a long history of medicinal use and an established role in plant-herbivore and predator-prey interactions. Cardenolides are classically studied steroidal defense mechanisms in chemical ecology and the coevolution of plants and herbivores. Although milkweed plants (Asclepias) produce up to 200 structurally different cardenolides, all compounds appear to share the same well-characterized mode of action, inhibition of the ubiquitous Na+/K+ ATPase in animal cells.
The authors focused on the study of compounds contained in A. curassavica seeds in order to describe their structure, concentrations and activity. HPLC-HRMS was used to isolate the compounds. The structures of the metabolites were determined based on data obtained from nmr measurements. The inhibitory capacity of the compounds was tested against porcine NKA. An in vitro assay was used to determine the IC50 of each compound and ouabain as a reference.
The work showed that the glycosylation of the compound leading to higher polarity is directly responsible for the decrease in the toxic potential of cardenolides. In addition, the structure of the glycosyl substituents and the degree of oxidation at position 19 (alcohol vs. aldehyde) in coroglaucigenin-type cardenolides may also affect the inhibitory capacity.
The work in the sense of performing the experiments and interpreting the results does not raise any major objections.
Comments
This sentence "Fourteen compounds have been previously described as seed constituents, several of which have only been found in the seeds" is imprecise what seeds the authors mean?
What is the opinion of the authors on the possibility of using their results, e.g. in medicine and cosmetics?
Author Response
We have carefully considered the reviewer´s suggestions, and produced a detailed description of how we have revised our manuscript (numbering of the comments is ours).
R2.1. This sentence "Fourteen compounds have been previously described as seed constituents, several of which have only been found in the seeds" is imprecise what seeds the authors mean?
We have revised the sentence to improve it’s clarity. This now reads (line 76 - 79): “In that study fourteen compounds were isolated, several of which have only been found in the seeds A. curassavica, in contrast to some cardenolides which are present in foliage, latex, and roots [45]”.
R2.2. What is the opinion of the authors on the possibility of using their results, e.g. in medicine and cosmetics?
Cardiac glycosides at the proper dose are used to treat congestive heart failure. They have strong anticancer properties, and the first generation of cardiac glycoside-based anticancer drugs are currently in clinical trials. Cardenolides are also being explored for the treatment of some tropical diseases. Because we work within an ecological framework, it is beyond the scope of this paper to discuss the compound’s potential use in medicine.

Reviewer 3 Report
The manuscript deals with the isolation and characterization of four new cardenolides from seeds of Asclepias curassavica. The main aim of this paper is to report the new compounds. However, the purity of new compounds 1, 2, 3, and 4 is compromised. Authors should describe the reason for the low purity of compounds after using different chromatographic techniques for the purification. Why is it so challenging to get these compounds with more than 95% purity? Authors should briefly describe their characteristics IR values in the result sections and also mention IR values in the methodology section. Please provide IR spectra of new compounds in supporting information. Please also include log ε values for all reported new compounds. Also mention the ratio of ACN/H2O.
The following comments and suggestions should be considered by the authors:
Abstract: Please briefly describe the objective of the study.
L11: ‘are’ is repeated twice in the sentence.
L81,82: Could you please briefly describe the cardenolide characteristics used for the putative structural assignment?
L96: Please italicize Asclepias curassavica throughout the manuscript.
L97: incomplete sentence?
L99: Could you please indicate rings A and B in the chemical structure (Figure 1)?
L111: 1H-13C HMBC?
NMR table: Please mention the type of carbons (C, CH2, CH3), the multiplicity of protons including overlapped, and mention if the coupling constant cannot be determined due to the second-order effect.
bd=broad doublet? If it is a broad doublet, please write brd?
L161: reversed-phase? Please use the reversed-phase throughout the manuscript.
L221: MeOH-d3
Please provide the residual solvent signals for D2O and MeOH-d3 used as internal standards.
L236-242: Could you please provide the LC-MS/MS conditions used for the analysis such as ion spray voltage, sheath gas, auxiliary gas, capillary temperature, etc?
Please italicize m/z throughout the manuscript.
L229-231: min?
L253: Could you please mention the flow rate and wavelength used for the UV detection?
L276: Please briefly describe the HPLC condition used (column, manufacturer, dimension in mm, flow rate mL/min, and UV detection at nm).
L286: (c 0.14, H2O). Use this format.
References: Please carefully check all references and use the ACS style.
No need to provide an issue number.
Please italicize the volume.
L401: provide volume and page number.
L405: use the abbreviation of the journal.
Author Response
We have carefully considered the reviewer’s suggestions, and produced a detailed description of how we have revised our manuscript (numbering of the comments is ours).
R3.1. The purity of new compounds 1, 2, 3, and 4 is compromised. Authors should describe the reason for the low purity of compounds after using different chromatographic techniques for the purification. Why is it so challenging to get these compounds with more than 95% purity?
The lower purity levels of compounds 1-4 was due to the persistent co-elution with several minor compounds. We have added this information on 294-295.
R3.2. Authors should briefly describe their characteristics IR values in the result sections and also mention IR values in the methodology section. Please provide IR spectra of new compounds in supporting information. Please also include log ε values for all reported new compounds. Also mention the ratio of ACN/H2O.
We are unable to provide IR data because we do not have an IR spectrometer in our institution. The new compounds were fully characterized by HRMS, NMR, and optical rotation measurements. This characterization leaves to our knowledge no open questions regarding the chemical structures. IR data would certainly confirm the presence of important functional groups, e.g. the unsaturated lactone ring, but they will not add substantial information regarding the structures reported. It is true that institutions less well equipped with modern analytical machines sometimes have only access to IR data. The
specialists concerned with the structure elucidation in such case will however compare their data with internet-accessible databases like Wiley’s SpectraBase where, for example, the IR spectrum of digitoxin is freely available.
R3.3. Abstract: Please briefly describe the objective of the study.
We have added a description of the objectives to the abstract (lines 14-17): “Milkweed seeds are eaten by specialist lygaeid bugs, which are the most cardenolide-tolerant insect known. These insects likely impose natural selection for the repeated derivatisation of cardenolides. A first step in investigating this hypothesis is to conduct a phytochemical profiling of the cardenolides in the seeds.”
R3.4. L11: ‘are’ is repeated twice in the sentence.
Duplicate deleted.
R3.5. L81,82: Could you please briefly describe the cardenolide characteristics used for the putative structural assignment?
We have added the description of the cardenolide characteristics on lines 86-89 “The cardenolide characteristics used for identification were the neutral loss that correspond to sugars in cardenolides (e.g. 162.05 Da- possible glucose, 146.05 Da-possible methyl allose). The spectrum must also contain high intensity fragments between m/z 353.20-391.25 tentatively corresponding to an unsaturated triterpene [47,48]”
R3.6. L96: Please italicize Asclepias curassavica throughout the manuscript Done.
R3.7. L97: incomplete sentence?
We have fixed the sentence structures (line 102-106)
“The molecular structures of Asclepias cardenolides have been intensively studied in the past [52–54]. Based on x-ray analysis, their principle structures have been determined: the triterpene scaffold of Asclepias cardenolides has the common feature of an -orientation of the methine proton at C-5, therefore the rings A and B of the scaffold are trans-fused [55].”
R3.8. L99: Could you please indicate rings A and B in the chemical structure (Figure 1)?
Figure 1 now includes this annotation.
R3.9. L111: 1H-13C HMBC?
Corrected.
R3.10. NMR table: Please mention the type of carbons (C, CH2, CH3), the multiplicity of protons including overlapped, and mention if the coupling constant cannot be determined due to the second-order effect. .bd=broad doublet? If it is a broad doublet, please write brd?
NMR table was revised and changed to include all multiplicities according to ACS guidelines and type of carbons.
R3.11. L161: reversed-phase? Please use the reversed-phase throughout the manuscript.
This was revised and changed across the manuscript.
R3.12. L221: MeOH-d3
Fixed in line 227.
R3.13. Please provide the residual solvent signals for D2O and MeOH-d3 used as internal standards.
Included in lines 227-229 “Samples were measured in MeOH-d3 (99.5%) or D2O (99.9%), depending on solubility of the compounds, residual solvent signals used as standard were δ 4.79 ppm and 3.31 ppm respectively.”
R3.14. L236-242: Could you please provide the LC-MS/MS conditions used for the analysis such as ion spray voltage, sheath gas, auxiliary gas, capillary temperature, etc?
We provide these conditions on lines 244-248 “Electrospray ionization (ESI) in positive ion mode was used for the analysis in full scan and auto MS/MS modes, scanning masses from m/z 50–1300. Capillary voltage set at 4500 V, nebulizer pressure gas at 1.8 bar, drying gas temperature at 220 OC and drying gas flow set at 9.0 L/min. Sodium formate adducts were used for internal calibration with HPC mode”
modeR3.15. Please italicize m/z throughout the manuscript
Done.
R3.16. L229-231: min?
Done.
R3.17. L253: Could you please mention the flow rate and wavelength used for the UV detection?
Fixed at lines 253-255 “A linear gradient, using a mobile phase consisting of MeOH (supplied with 0.1% FA, Carl Roth GmbH) and water (0.1% FA) was used, with a flow rate of 50 mL/min, and UV detection carried out at 218 nm”.
R3.18. L276: Please briefly describe the HPLC condition used (column, manufacturer, dimension in mm, flow rate mL/min, and UV detection at nm).
HPLC conditions are described in section 3.1. General Experimental Procedures.
Preparative HPLC experiments were made with the conditions described in lines 259-264. “Semipreparative HPLC separations were carried out on a Shimadzu Prominence HPLC System, consisting of an autosampler SIL-20AC, gradient pump LC-20AT, UV-Vis detector SPD-20A programed for detection at 220 nm, and a fraction collector FRC-10A. For separations, isocratic elutions with MeOH-water mixtures were per-formed at flow rate of 0.8 mL/min. A C-18 Nucleodur Isis column (4.6 x 250 mm, 5 μm particle size, from Macherey-Nagel, Düren, Germany) was used”.
R3.19. L286: (c 0.14, H2O). Use this format.
Done.
R3.20. References: Please carefully check all references and use the ACS style. No need to provide an issue number. Please italicize the volume. L401: provide volume and page number. L405: use the abbreviation of the journal.
Done

Round 2
Reviewer 3 Report
There are still several typographical errors. The authors should carefully revise the manuscript.
L131: Please italicize dH and dC.
L134: Can you italicize J?
Please mention the field strength of NMR for proton and carbon.
Mention the residual solvent signals for both carbon and proton in the methodology.
Please include log ε values for all reported new compounds. Also mention the ratio of ACN/H2O.
L276: 11.0 g?
Please carefully check the references and modify them according to ACS style. L416: volume? L426, 462, 549: year of publication should be bold. L502: 119, e2205073119?
Author Response
R3.1 There are still several typographical errors. The authors should carefully revise the manuscript This was revised and fixed across the manuscript.
R3.2 L131: Please italicize dH and dC. Done.
R3.3 L134: Can you italicize J? Done.
R3.4 Please mention the field strength of NMR for proton and carbon. This information was added at the Table 1 tittle Line 138. And stated in the Material and methods “Field strengths of 1H (500.13 MHz)/13C (125.76 MHz): 11.747 T.” line 227.
R3.5 Mention the residual solvent signals for both carbon and proton in the methodology. “For compounds measured in MeOH- d3 the residual solvent signals were δH 3.31/ δC 49.15. For measurements in D2O all chemical shifts were left uncorrected after carefully tuning and matching the NMR instrument.” Added in line 231.
R3.6 Please include log ε values for all reported new compounds. The reporting of log ε is common when one records a UV spectrum in a dedicated UV spectrometer, but not for HPLC analyses with a DAD detector as it is in our capacity. Therefore, accuracy on ε coefficient is not within our possibilities at the moment. We can only infer that the log ε for cardenolides of the same type is about 4.22 at 217 nm. The new compounds were fully characterized by HRMS, NMR, and optical rotation measurements. This characterization leaves to our knowledge no open questions regarding the chemical structures.
R3.7 Also mention the ratio of ACN/H2O. The ratio were the new compounds were isolated using preparative HPLC is described in lines 295-299 as it follows “F9 (77.7 mg) separated with a mobile phase of ACN/H2O (16:84) gave compound 1 (3 mg, 75% purity). F6-8 (86.7, 39.6, and 48.6 mg, respectively) were separated in a similar way (ACN/H2O isocratic mobile phase: 12:88, 14:86, and 15:85, respectively) and gave compounds 2 (3.7 mg, 91% purity), 3 (1.1 mg, 58% purity), 4 (2.5 mg, 48% purity), and 16α-hydroxycalotropin (10, 4.4 mg, 58% purity).”
R3.8 L276: 11.0 g? Fixed.
R3.9 Please carefully check the references and modify them according to ACS style. L416: volume? L426, 462, 549: year of publication should be bold. L502: 119, e2205073119? Done.